# ANN Prediction Model of Concrete Fatigue Life Based on GRW-DBA Data Augmentation

**Jinna Shi** [1,2,3], **Wenxiu Zhang** [1] **and Yanru Zhao** [1,2,3,*]

1   School of Civil Engineering, Inner Mongolia University of Technology, Hohhot 010051, China
2   Inner Mongolia Autonomous Region Building Inspection, Appraisal and Safety Assessment Engineering Technology Research Center, Hohhot 010051, China
3   Key Laboratory of Civil Engineering Structure and Mechanics, Inner Mongolia University of Technology, Hohhot 010051, China
*   Correspondence: zhaoyanru710523@126.com

**Abstract:** In order to improve the prediction accuracy of the machine learning model for concrete fatigue life using small datasets, a group calculation and random weight dynamic time warping barycentric averaging (GRW-DBA) data augmentation method is proposed. First, 27 sets of real experimental data were augmented by 10 times, 20 times, 50 times, 100 times, 200 times, 500 times, and 1000 times, respectively, using the GRW-DBA method, and the optimal factor was determined by comparing the model's training time and prediction accuracy under different augmentation multiples. Then, a concrete fatigue life prediction model was established based on artificial neural network (ANN), and the hyperparameters of the model were determined through experiments. Finally, comparisons were made with data augmentation methods such as generative adversarial network (GAN) and regression prediction models such as support vector machine (SVM), and the generalization of the method was verified using another fatigue life dataset collected on the Internet. The result shows that the GRW-DBA algorithm can significantly improve the prediction accuracy of the ANN model when using small datasets (the $R^2$ index increased by 20.1% compared with the blank control, reaching 98.6%), and this accuracy improvement is also verified in different data distributions. Finally, a graphical user interface is created based on the developed model to facilitate application in engineering.

**Keywords:** artificial neural network; data augmentation; fatigue life; predictive model; small datasets

## 1. Introduction

Concrete is one of the most widely used materials in the construction industry, and its fatigue damage seriously affects the safety of structures such as crane girders [1], pavements [2], and bridges [3]. Accurate prediction of fatigue life is of great engineering significance for the safety of concrete structures [4,5].

The fatigue life of concrete is affected by many complex factors, such as material, load, and environment. These factors make it difficult to establish an accurate concrete fatigue life prediction model. The classical methods for predicting the fatigue life of concrete mainly include fracture mechanics, continuum damage mechanics, and energy methods [6,7]. However, due to the influence of empirical parameters and specific materials, these mathematical and physical modeling methods have low prediction accuracy and poor generalizability. At present, the application of artificial intelligence has developed in various research fields of machinery [8–10], medicine [11–14], and civil engineering [15]. Many scholars use support vector machine (SVM) [16], decision trees [17], random forests [18], and adaptive boosting (AdaBoost) [19] to predict the mechanical strength properties and mix proportions of concrete and obtain higher prediction accuracy than traditional models. However, there is still room for improvement in the prediction accuracy of these statistical

probability-based models in complex and highly uncertain data. In recent years, artificial neural network (ANN) has been widely used due to its powerful nonlinear fitting ability. ANN does not require functional assumptions and can learn complex nonlinear relationships through a data-driven training process, which is advantageous in fatigue life prediction research [20]. There are many studies on concrete fatigue life prediction using ANN, and higher prediction accuracy has been achieved in experiments [21,22]. Therefore, ANN is considered a suitable method to solve the problem of concrete fatigue life prediction with high uncertainty.

The high prediction accuracy of the ANN model largely depends on sufficient training data. However, the fatigue life of concrete is usually determined by experiments, and the available training data are limited. When the training set is too small, the model will overfit the distribution trend of the training set and cannot grasp the development trend of concrete fatigue life, resulting in an overfitting phenomenon [23]. Data augmentation is an effective tool to increase the number of training sets, which increases the amount of data used for model training by generating synthetic data to improve the prediction accuracy of the model [24]. Classic sequential data augmentation methods include temporal transformations, statistical generative models, and learning-based models [25]. Time-domain transformation methods mainly include sampling, slicing, flipping, etc., but it is not easy to confirm whether this method affects the sequence distribution [26]. Statistical generative models such as mixed autoregressive (MAR) [27], use statistical models to simulate the distribution of data, but they rely too much on the initial value. Once the initial value is disturbed, the data will be generated according to different conditional distributions.

Learning-based models, such as generative adversarial networks (GANs) [28,29], evolutionary search [30], etc., generate augmented data based on the exact fit of the generator to the distribution of the source data and are currently widely used in the image domain. Amyar et al. proposed a deep convolutional conditional generation adversarial network to generate MIP positron emission tomography (PET) images, which solved the problem of category imbalance and lack of data in medical imaging [29]. However, in augmentation of series data, such learning-based models perform unstably in too small datasets [31,32]. Given this, Fawaz et al. proposed an average data augmentation method based on dynamic time warping (DTW) distance [33] called DTW barycentric averaging (DBA) and obtained at least 60% of the two training sets in the UCR archive (containing 16 sets of data and 57 sets of data, respectively). The improved prediction accuracy demonstrates the effectiveness of the method for small datasets [34]. However, this method also has problems, such as the amplification result being easily affected by abnormal sequences and the amplification process being cumbersome.

Therefore, this paper proposes a group calculation and random weight dynamic time warping barycentric averaging (GRW-DBA) data augmentation algorithm and ANN model for concrete fatigue life prediction with small datasets through the following innovations.

(1) This paper proposes an optimized GRW-DBA data augmentation algorithm based on group computing and random weight mechanism. Compared with other algorithms such as classic DBA, the GRW-DBA algorithm has a simpler operation process, is not easily affected by outliers, and can obtain generated data that are more in line with the distribution of source data.

(2) We construct a prediction model based on a GRW-DBA data augmentation algorithm and ANN and develop a graphical user interface. Compared with classical mechanical methods, the model can significantly improve the prediction accuracy and at the same time facilitate engineering applications.

The paper is organized as follows. Section 2 shows the innovative data augmentation algorithm and the predictive model used. Section 3 describes the extended data-based modeling process and the experimental process of model hyperparameter selection. Section 4 describes the experimental validation. Section 5 provides a discussion of the results. Section 6 draws conclusions.

## 2. Methods

### 2.1. Group Random Weight DBA Algorithm

The GRW-DBA method proposed in this paper is based on the classic DBA method, which improves the grouping calculation and random weights and overcomes the shortcomings of the classic DBA method [34], which is easily affected by data outliers and cumbersome augmentation processes. The process of data augmentation by this algorithm will be introduced in detail below.

First, divide calculation groups. Different from the classic DBA algorithm that uses all data to perform calculations, the GRW-DBA algorithm divides $N$ sets of data into multiple computational groupings, such as any 2 as groups, any 3 as groups, any 4 as groups, ... , and all $N$ as groups. Each calculation group is independent of the other, and subsequent calculations are performed independently.

Second, determine the sequence weight. In each calculation group, the initial sequence $Q$ is first randomly selected and assigned a non-repeating random weight $x$. Then, the DTW distance [33] between the initial sequence $Q$ and any other sequence $P$ in the group is calculated. Suppose the two sequences $Q$ and $P$ are $Q = [q_1, q_2, q_3, \cdots, q_m]$ and $P = [p_1, p_2, p_3, \cdots, p_n]$, respectively. The distance between elements $q_i$ and $p_j$ in the sequences is calculated by the formula:

$$d(i, j) = (q_i - p_j)^2 \tag{1}$$

The distances between all corresponding elements of two sequences form an $m \times n$ distance matrix $M$; $M$ can be expressed as:

$$M = \begin{bmatrix} d(m,1) & d(m,2) & d(m,3) & \cdots & d(m,n) \\ \vdots & \vdots & \vdots & \cdots & \vdots \\ d(3,1) & d(3,2) & d(3,3) & \cdots & d(3,n) \\ d(2,1) & d(2,2) & d(2,3) & \cdots & d(2,n) \\ d(1,1) & d(1,2) & d(1,3) & \cdots & d(1,n) \end{bmatrix} \tag{2}$$

Taking $d(1, 1)$ as the starting point, select one of the elements which is above, right, and top right of the starting point. Then, based on the element, repeat the same steps in this way until $d(m, n)$ is reached. These selected elements form a path $R$ between sequences $Q$ and $P$, which can be denoted as $R = \{d(r_1), d(r_2), \cdots, d(r_s), \cdots, d(r_N)\}$, where $N$ denotes the total number of elements in the path, $r$ is the coordinate of the point on the path, i.e., $r_s = (i, j)$, and there are many of these paths $R$. However, there must exist an optimal path in all path spaces $R$ that minimizes $\sum_{s=1}^{N} d(r_s)$. Therefore, the DTW distance between sequences $Q$ and $P$ is:

$$\text{DTW}(Q, P) = \min(\sum_{s=1}^{N} d(r_s)) \tag{3}$$

where $d(r_s)$ is the path distance calculated between each corresponding value, and $N$ represents the number of sequence data. To solve for the minimum $\sum_{s=1}^{N} d(r_s)$ value, the cumulative distance matrix $D$ is calculated using the dynamic programming method and the corresponding distance $d$ between two points as:

$$D(i, j) = d(i, j) + \min\{D(i, j-1), D(i-1, j-1), D(i-1, j)\} \tag{4}$$

where $i = 1, 2, 3, \cdots, m$, and $j = 1, 2, 3, \cdots, n$. The last element $D(m, n)$ of $D$ is the final DTW distance, i.e.,

$$\text{DTW}(Q, P) = D(m, n) \tag{5}$$

By comparing the DTW distance between the initial sequence and other sequences in the calculation group, we find the two sequences with the smallest and second smallest DTW distances from the initial sequence and assign a weight of $0.3x$ to each of these two

sequences. Finally, the remaining $(1 - 1.6x)$ weights are equally distributed to the other remaining sequences. Since $x$, $0.3x$, and $1 - 1.6x$ represent the weight of each sequence, it is very important to ensure that all three values are between 0 and 1 according to the experience of the literature [35]. Therefore, it can be calculated that the value range of $x$ in this case is between 0 and 0.625. The generation of random weights adopts the mature Mersenne Twister algorithm [36] in the computer, which generates pseudo-random numbers by continuously calling the algorithm. After the judgment process, non-repeating values that meet the requirements are used as random weights and parameters.

We use this method in all calculation groups to calculate combinations of more than four sequences; due to the limitation on the number of sequences, combinations of two and three sequences do not satisfy the above weight distribution scheme, so these groups assign uniform weights to all sequences.

Third, we find the weighted average by group. First, in each computation group, the sequences are weighted and averaged according to the already determined weights to obtain an augmented set of sequences. Since the calculation combination of two sequences and three sequences keeps its calculation result unchanged every time it is repeated, the result is multiplied by a random factor between zero and one here as the final result. Then, we aggregate the extended sequences obtained from all calculation groups to obtain all augmented sequences for one iteration.

Finally, steps 2 and 3 are run iteratively until the amount of augmented data satisfies the requirement. Then, a certain amount of data is randomly selected from all augmented data to form an augmented dataset.

The flowchart of the GRW-DBA algorithm is shown in Figure 1. It can be seen from the algorithm flow that since this method uses group calculations instead of all average calculations of the classical DBA method, it can not only reduce the impact of abnormal data but also obtain a large number of extended sequences in one algorithm iteration. At the same time, the GRW-DBA method uses random weights instead of fixed weights, so the augmented sequences obtained in each iteration are not repeated, and only a few simple iterations are required to obtain the required number of augmented sequences. Taking Figure 1b as an example, in the augmentation experiment of 7 sets of data, 120 sets of augmented sequences can be obtained in one iteration using the GRW-DBA method.

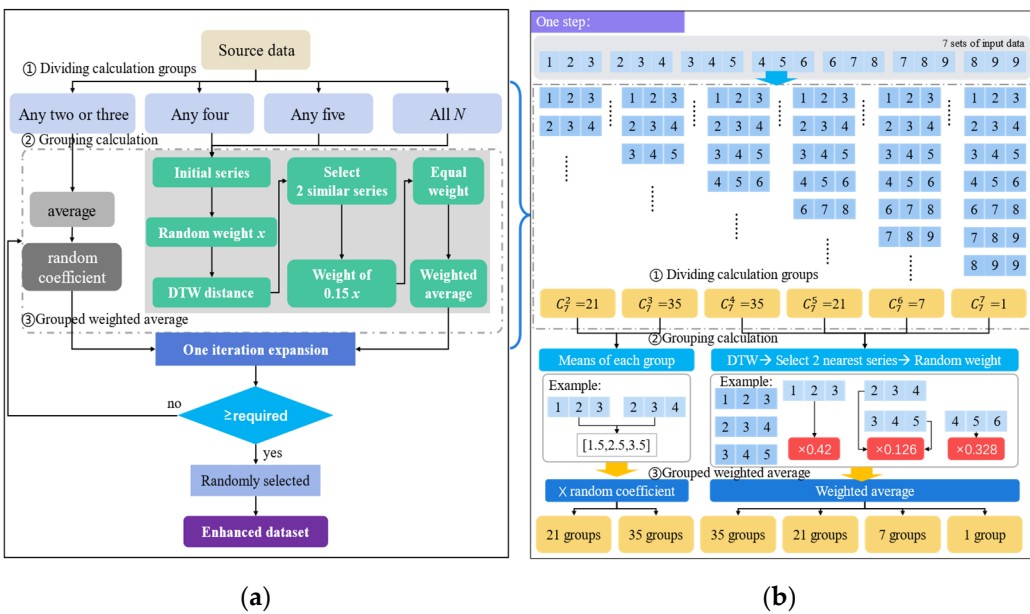

(**a**)　　　　　　　　　　　　　　　　　　　　　　(**b**)

**Figure 1.** GRW-DBA algorithm flowchart: (**a**) algorithm flowchart; (**b**) schematic diagram of one-step data augmentation process.

### 2.2. Artificial Neural Networks

The ANN model consists of an input layer, a hidden layer, and an output layer [37]. The number of neural units in the input layer is consistent with the number of input variables, and the whole structure parameters related to fatigue life are set as the input layer in this paper. The number of neural units in the output layer is consistent with the number of output variables. The fatigue life value is used as the output layer. The overall structure of the model is shown in Figure 2.

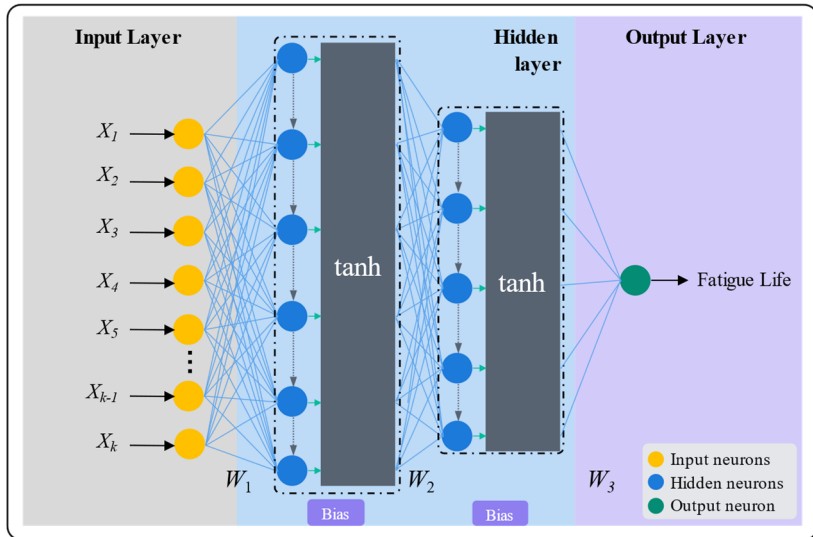

**Figure 2.** Schematic diagram of ANN model structure.

The neurons within the model are connected by weights and biases, while the nonlinear activation function (tanh) is used to complete the fitting of the nonlinear data. The weight of a neuron represents the importance of the output value of the neuron in the final prediction, while the bias is responsible for the translation of the neuron output value. Thus, the output value of a layer of neurons can be expressed as Equation (6).

$$Y = \tanh(w_i x_i + b_l) \tag{6}$$

where $Y$ denotes the output of a layer of neurons, $x_i$ denotes the value of each neuron, and tanh is the hyperbolic activation function, which is calculated as:

$$\tanh(x) = (1 - e^{-x})/(1 + e^{-x}). \tag{7}$$

In addition, $w_i$ denotes the weight of the neuron and $b_l$ denotes the bias of the layer. Proper weights and biases can make the model perform well not only on training data but also on maintaining good prediction accuracy on validation data [38].

## 3. Fatigue Life Prediction Modeling

### 3.1. Datasets

Dataset 1: Experimental data used in model building and experiments to determine hyperparameters were obtained from the team's research published in *Structural Concrete* [39]. The dataset contains a total of 27 sets of data, using 24 parameters that characterize the pore structure of concrete as the input variables of the model, and the concrete fatigue life as the output variable.

Dataset 2: To verify the performance of the model in different data distributions, in Section 4.3, the dataset obtained in the paper [40] is used for experiments. The dataset contains 28 sets of data in total, 6 variables related to fatigue life are used as the input

variable, and concrete fatigue life is used as the output variable. A detailed explanation of the input variables for the two datasets is given in Table 1.

**Table 1.** Explanation of the meaning of the input variables in the datasets.

| Datasets | Number | Input Variables | Explanation |
|---|---|---|---|
| Dataset 1 | 1 | *P1%* | Pore size of 0.1–27.98 nm |
| | 2 | *P2%* | Pore size 27.98–524.26 nm |
| | 3 | *P3%* | Pore size 524.26–6463.30 nm |
| | 4 | *PM1%* | Porosity of pore size corresponding to highest peak of *P1* part of the curve |
| | 5 | *PM3%* | Porosity of pore size corresponding to highest peak of *P3* part of the curve |
| | 6 | *S1* | Rate of change of *P1* pore size |
| | 7 | *S3* | *P3* rate of pore size change |
| | 8 | *Dna* | *P1* pore fractal dimension |
| | 9 | *Dnb* | *P3* pore fractal dimension |
| | 10 | *P1Q%* | *P1* part of the 0.1–7.5 nm pore size porosity |
| | 11 | *P1b%* | *P1* part less than 5 nm pore size porosity |
| | 12 | *P1h* | *P1* part 5–27.98 nm pore size porosity |
| | 13 | *Sz* | Pore size integrated change rate |
| | 14 | *Cz* | Pore structure complexity factor |
| | 15 | *Large capillaries* | Pore size of 50–10,000 nm |
| | 16 | *Small capillaries* | Pore size 10–50 nm |
| | 17 | *Inter-colloidal pores* | Pore size 2.5–10 nm |
| | 18 | *Micropores* | Pore size 0.5–2.5 nm |
| | 19 | *Interlayer pores* | Aperture size is less than 0.5 nm |
| | 20 | *Non-harmful pores* | Pore size is less than 20 nm |
| | 21 | *Less harmful pores* | Pore size is 20–100 nm |
| | 22 | *Harmful pores* | Pore size is 100–200 nm |
| | 23 | *Multi-harmful holes* | Pore size is greater than 200 nm |
| | 24 | $P_{total}$ | Total pore size |
| Dataset 2 | 1 | *fc* | Compressive strength of concrete |
| | 2 | *h/w* | Height-to-width ratio |
| | 3 | *Shape* | Shape of the test specimens |
| | 4 | *Smax* | Maximum stress level |
| | 5 | *R* | Minimum stress to maximum stress ratio |
| | 6 | *f(HZ)* | Loading frequency |

### 3.2. Evaluation Indexes

In this paper, three evaluation indexes are used to evaluate the prediction accuracy of the model. They are correlation coefficient ($R^2$), root mean square error (RMSE), and mean absolute error (MAE). Among them, $R^2$ indicates the similarity between the predicted and actual values given by the model, and the closer the value is to one, the closer the predicted value is to the actual value; RMSE and MAE are the average error magnitude between the predicted and actual values of fatigue life derived from the model using the two calculation methods, and the smaller the value is, the smaller the error magnitude between the predicted and actual values is. Therefore, the closer the values of RMSE and MAE are to zero and the closer the value of $R^2$ is to one, the higher the prediction accuracy of the model. The three evaluation indicators are defined by Equations (8)–(10).

$$R^2 = \frac{\sum\limits_{i=1}^{n} (y_i - \overline{y}_i)(\hat{y}_i - \overline{\hat{y}}_i)}{\sqrt{\sum\limits_{i=1}^{n} (y_i - \overline{y}_i)^2} \sqrt{\sum\limits_{i=1}^{n} (\hat{y}_i - \overline{\hat{y}}_i)^2}} \tag{8}$$

$$\text{RMSE} = \sqrt{\frac{1}{n}\sum_{i=1}^{n}(y_i - \hat{y}_i)^2} \tag{9}$$

$$\text{MAE} = \frac{1}{n}\sum_{i=1}^{n}|(y_i - \hat{y}_i)| \tag{10}$$

where $n$ is the total number of values, $\hat{y}$ denotes the actual value, $y$ is the predicted result calculated by the model, $\overline{\hat{y}}$ is the average of the actual value, and $\overline{y}$ denotes the average of the predicted result.

### 3.3. Modeling Process

Step 1: Prepare the source dataset. The source dataset is constructed based on dataset 1 in Section 3.1. The dataset contains a total of 27 sets of data, and each set of data consists of fatigue life values (output variables) and corresponding 24 pore structure parameters (input variables). At the same time, the dataset is divided into 70% training data and 30% test data to verify the effect of data enhancement as a blank control.

Step 2: Prepare the augmented datasets. According to the experience of the literature [34], we use the GRW-DBA method described in Section 2.1 to perform data augmentation on the source dataset, and we enhance the 27 sets of data of the source data to 10 times, 20 times, 50 times, 100 times, 200 times, 500 times, and 1000 times. These augmented datasets are all split into 70% training data and 30% validation data.

Step 3: Determine hyperparameters. First, to determine the optimal factor of augmentation, the effect of augmented data with 10, 20, 50, 100, 200, 500, and 1000 times are compared. Then, in order to make the model achieve fast and accurate prediction, it is necessary to select the appropriate number of training iterations, the number of hidden layers, and the learning rate constant $\alpha$. With reference to the literature [41], the number of iterations was chosen to be 100, 500, 800, 1000, and 1200. The number of hidden layers is generally chosen from small to large and should not be too large [42], so they are chosen as follows: 1, 2, 3, and 4. In practice, the learning rate constant is generally taken to be $10^{-n}$, with $n$ being a positive integer [43]. Thus, the rate constants $\alpha$ are chosen as: 0.01, 0.001, 0.0001, and 0.00001. The modeling flowchart is shown in Figure 3. In the experiments, Origin software was used to generate and export the images.

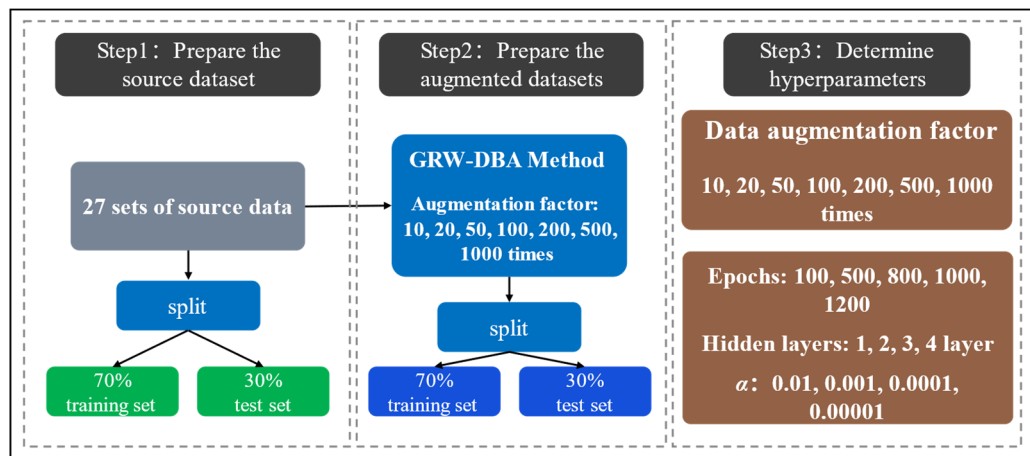

**Figure 3.** Schematic diagram of the modeling process of the prediction model based on GRW-DBA and ANN.

### 3.3.1. Determine the Data Augmentation Factor

Experiments were conducted based on Dataset 1, where the hyperparameters of the ANN model are shown in Table 2. Three evaluation indexes including RMSE, MAE and $R^2$ were used to evaluate the prediction effect.

**Table 2.** Hyperparameters used by the ANN model.

| Hyperparameters | Values |
|---|---|
| Number of neurons in the input layer | 24 |
| Hidden layers | 80, 40 |
| Learning rate | 0.001 |
| Activation function | tanh |
| Iteration times | 1000 |

Figure 4 shows the performance and training time of the model in different augmented data amounts. From the results of the evaluation metrics of the comparative experiments, it is found that when the model is trained and tested using real experimental data or a smaller number of multiples of the augmented data, the prediction accuracy of the model is not high due to the small amount. This result proves that the model tends to be overfitted when the amount of data is small [23]. With the increase of the augmentation multiplier, the model could fully exploit the hidden features of the input data while avoiding the overfitting phenomenon. The prediction accuracy shows an overall increasing trend.

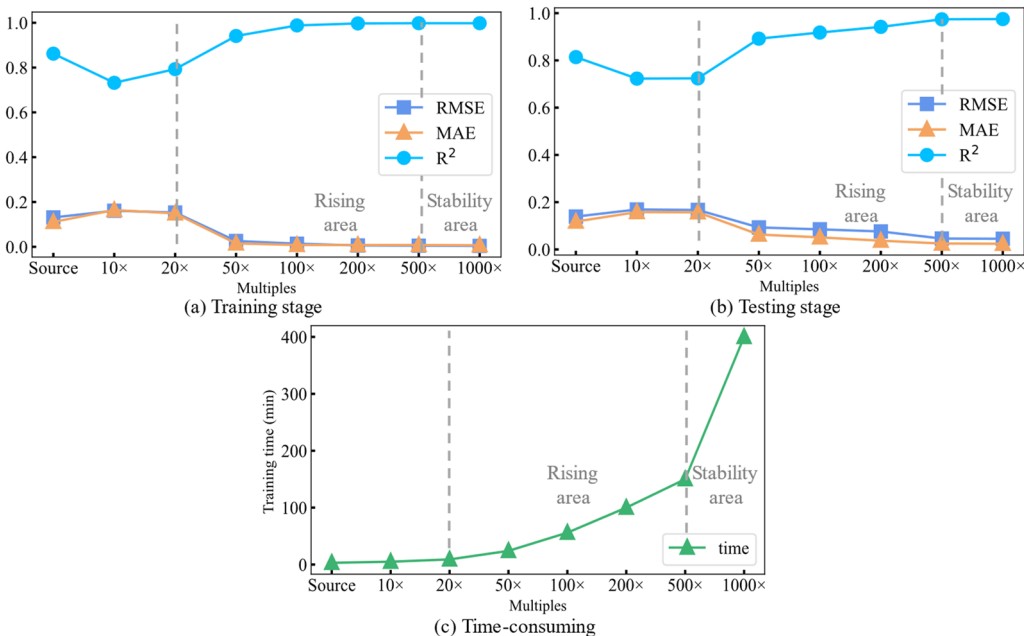

**Figure 4.** Comparison of prediction accuracy and training time of models with different augmentation multiples: (**a**) stage of training; (**b**) stage of testing; (**c**) time-consuming.

However, the increased amount of data also brings the problem of time consumption for model training. As shown in Figure 4c, the time used for model training is significantly longer when using 1000 times the augmented dataset compared to 500 times the data. Furthermore, the improvement in prediction accuracy is low when combined with Figure 4a,b. Therefore, the fatigue life prediction model is trained using 500 times (13,500 sets) of augmented data.

3.3.2. Determine the Hyperparameters of the ANN Model

1.  Iteration times:

In order to test the prediction performance for 100, 500, 800, 1000, and 1200 iterations, the data with 500 times augmented were used for training, the source data were used for testing. The training of the model is essentially the process of adjusting the connection weights between neural units. The initial values of these weights are randomized. With each iteration of the model, the optimization algorithm of the model adjusts the values

of the weights backward according to the difference between the output value given by the model and the actual value during the iteration, finally minimizing the difference [44]. Therefore, the appropriate number of iterations is a key factor that affects the prediction accuracy of the model. In the experiment, except for the number of iterations, other model parameters remain unchanged.

Figure 5 shows the results of model prediction accuracy evaluation for different iteration times. To clearly characterize the resulting data of 10 independent repetitions of the experiment, the successive results of the test set are represented as box plots. The box plot shows not only the trend of the evaluation indexes with the number of iterations but also the distribution of all the results in the 10 repeated experiments more effectively. The horizontal line in the middle of the box plot represents the median value of the 10 data, and the outermost horizontal line represents the maximum and minimum values of the data.

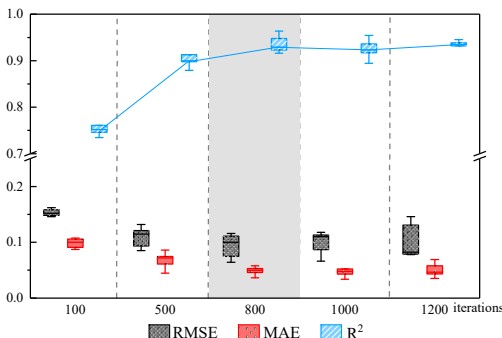

**Figure 5.** Evaluation indexes of prediction accuracy of ANN model under different iteration times.

From the results of the iterative experiments, we found that the RMSE and MAE evaluation indexes of the test set of the model both gradually decrease, and the $R^2$ index gradually increases as the number of iterations increases, which indicates that the prediction accuracy of the model is gradually improving as the number of iterations increases. It is proved that a higher number of iterations has a positive impact on the model performance. However, after reaching 800 iterations, the indexes of the test set fluctuated steadily. It indicated that the model has achieved good results. The higher the number of iterations, the longer it takes. Therefore, 800 is the best number of iterations.

2.    Hidden layers:

As the number of hidden layers increases, more weights and biases are involved in the nonlinear computation of the data inside the model. Then, it can better fit the data nonlinearly. However, as the model structure becomes more complex, the training time of the model also increases. What is more, an overly complex model causes overfitting, and too many hidden layers also cause the optimization of the model to stagnate at a local optimum point [45], which cannot further improve the accuracy of the model. Therefore, a reasonable number of hidden layers is crucial for prediction effectiveness. In order to select an optimal choice of the number of hidden layers, all the model parameters were kept constant except for the four hidden layers which were set to 1, 2, 3, and 4. All the 500 times augmented data were used as training data, real experimental data were used as test data, and the number of iterations was fixed to 800 times.

Figure 6 shows the trends of RMSE and MAE evaluation indexes when the hidden layers are different. The closer to 0, the higher the model accuracy is. It can be seen from Figure 6 that when the number of hidden layers is 1 to 3, the evaluation index value of the model gradually decreases with the increase of the number of hidden layers, which shows that as the complexity of the model increases, the prediction accuracy of the model also improves. However, when the number of hidden layers exceeds three, both the training and testing of the model exhibit a decrease in accuracy. This indicates that three hidden layers are more suitable for fatigue life prediction under this experimental condition.

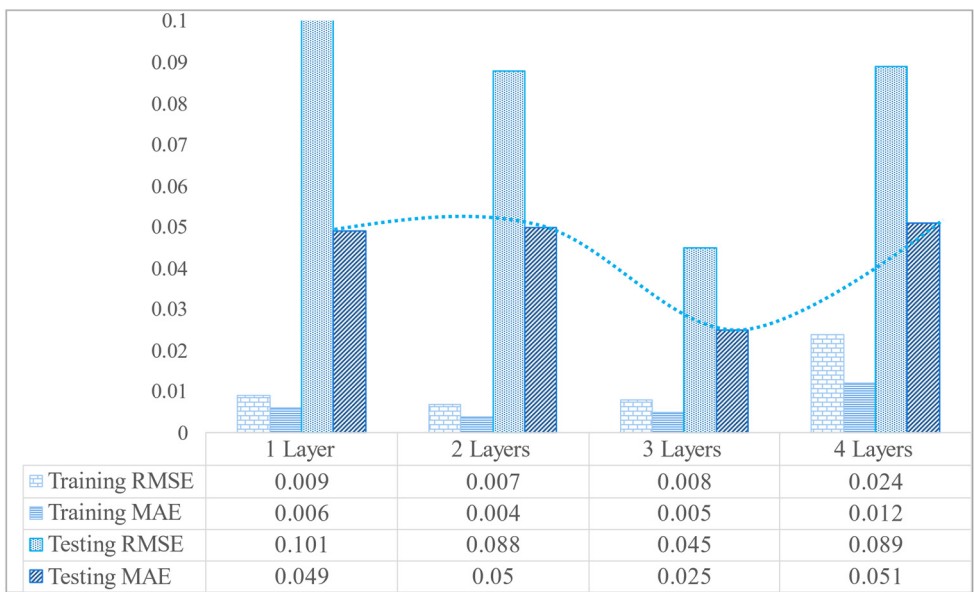

**Figure 6.** Evaluation indexes of prediction accuracy of ANN model under different hidden layers.

3.    Learning rate:

An appropriate learning rate is crucial to the training effect of the model. When the rate is too large, the model parameters may repeatedly exceed the minimum value of the objective function, resulting in suboptimal parameters. When the rate is too small, the model parameters may not converge to the optimal value within a limited time, thus affecting the prediction accuracy of the model [46].

In this experiment, four different learning rate constants were used as follows: 0.01, 0.001, 0.0001, and 0.00001. In the comparison test, 500 times augmented data were used as training data, source data were used as test data, the number of training iterations was 800, the number of hidden layers was 3, and other parameters were kept constant.

Figure 7 shows the trend of prediction accuracy at four different learning rates. As shown in the figure, the prediction accuracy of the model gradually increases when the learning rate is selected as 0.01, 0.001, and 0.0001 and decreases significantly when the learning rate is reduced to 0.00001. This shows that when the learning rate is set to 0.00001, the model cannot find the optimal value in time due to the too small learning rate. Therefore, when the learning rate is 0.0001, the accuracy of the model is the highest.

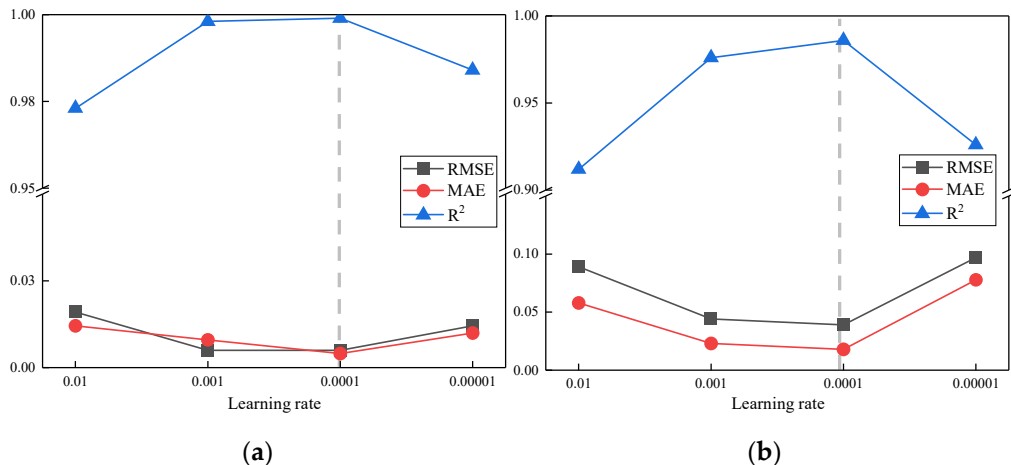

**Figure 7.** The trend of model prediction accuracy at different learning rates: (**a**) results of the training stage; (**b**) results of the testing stage.

## 4. Experimental Verification

A comparative experiment was designed to test the effectiveness of the GRW-DBA data augmentation method proposed in this paper combined with the ANN model for prediction. According to the augmentation multiple and model hyperparameters determined in Section 3, in the comparative experiment, the augmentation multiple of the source data was set to 500 times, the number of iteration was set to 800 times, there were 3 hidden layers, and the learning rate of the model was 0.0001. The evaluation indexes of prediction accuracy adopted RMSE, MAE, and $R^2$.

### 4.1. Validation of Data Augmentation Effects

In order to verify the effectiveness of the GRW-DBA data augmentation method proposed in this paper, the three classic methods of sampling [47], GAN [28], DBA [34], and our method were used to augment dataset 1, and the augmented data were used to train the ANN model. As a test set, the prediction accuracy of the four trained models was evaluated. At the same time, the ANN model using 70% of the source data as the training set was used as a blank control. Figure 8 shows the model prediction accuracy obtained by the four augmentation methods.

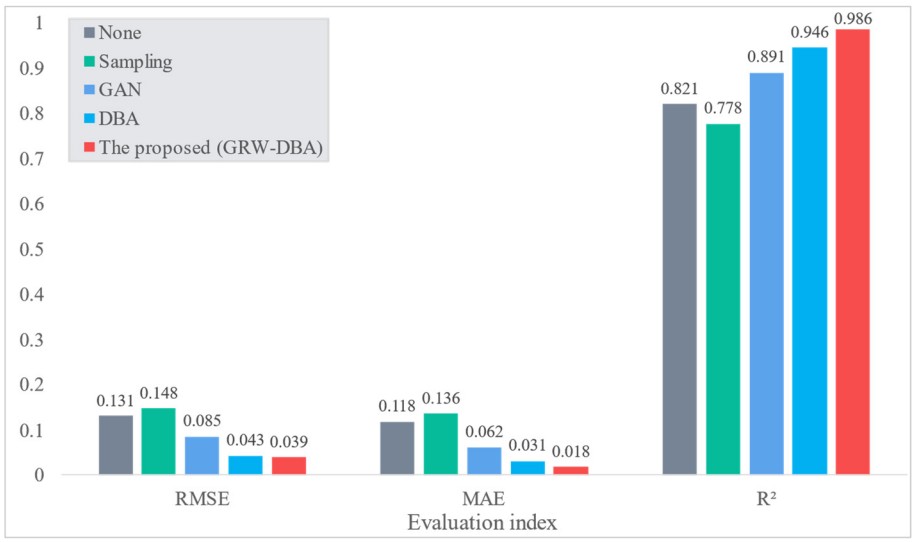

**Figure 8.** Comparison of the prediction accuracy of models trained on augmented data obtained using the GRW-DBA method proposed in this paper and other classical methods.

The smaller the RMSE and MAE index values, the larger the $R^2$ index value, and the better the prediction effect of the model. As shown in Figure 8, the prediction accuracy of the model trained using the augmented data obtained by GRW-DBA proposed in this paper is better than that of GAN [28] and other comparable models. At the same time, compared with the blank group obtained by using the source data training, the prediction accuracy of the model is greatly improved. It shows that the augmented data obtained by the GRW-DBA method is more in line with the distribution of the source data, and at the same time, the prediction accuracy of the ANN model under the condition of small data volume is effectively improved.

### 4.2. Validation of Predictive Models

In order to verify the effectiveness of the ANN prediction model used in this paper, the ANN model was compared with three classic regression prediction models: logistic regression (LR) [48], SVM [16], and AdaBoost [19]. The models are trained with the augmented data of GRW-DBA and tested with the source dataset.

Figure 9 shows that the $R^2$ of the ANN model combined with the GRW-DBA method reaches 0.986, which is higher than other models. Figure 10 shows the comparison of the

predicted value of the four comparison models based on 500 times augmented data. From Figure 10, it can be found that compared with the classical prediction model, the ANN model combined with the GRW-DBA data augmentation method can better predict the fatigue life of concrete.

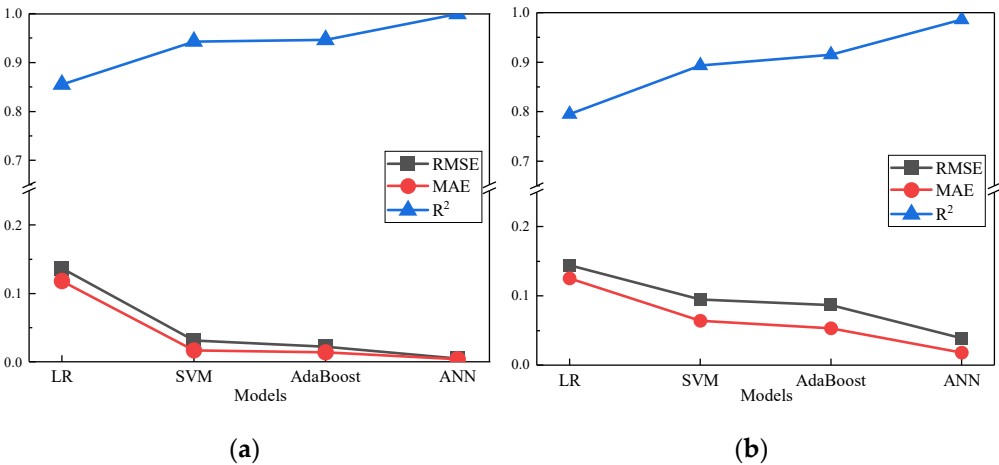

(**a**)　　　　　　　　　　(**b**)

**Figure 9.** The prediction accuracy evaluation indexes of four comparison models: (**a**) results of the training stage; (**b**) results of the testing stage.

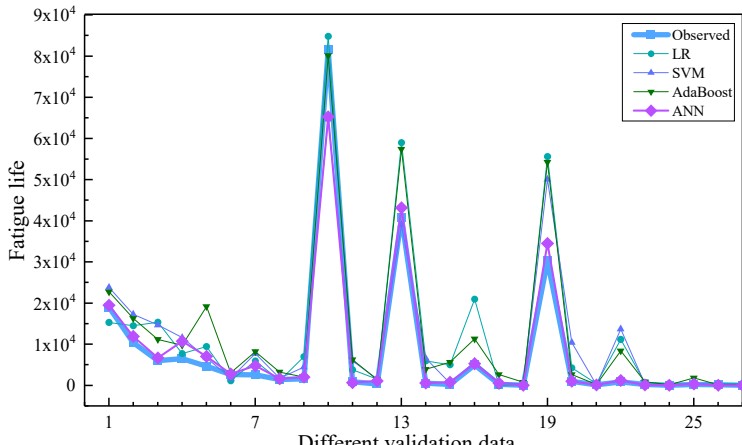

**Figure 10.** Predicted value of the four comparative models.

### 4.3. Verification of Generalization

To verify the generalization of the model, we used dataset 2 mentioned in Section 3.1 to conduct experiments. The settings of data augmentation factor and hyperparameters remained unchanged. The comparison model still used three classic regression prediction models: LR [48], SVM [16], and AdaBoost [19]. The data obtained by the augmentation of the GRW-DBA method was used as the training set, and the source data were used as the test set. The differences between the predicted values and observed values given by the four comparison models are shown in Figure 11. The *x*-axis represents the observed value, and the *y*-axis represents the predicted value. The closer the predicted value given by the model is to the observed value, the closer the points in the graph are to the *y* = *x* line. It can be seen that in the new data distribution, the prediction method of GRW-DBA combined with the ANN used in this paper still gives the result closest to the observed value, which proves the generalization of the proposed method.

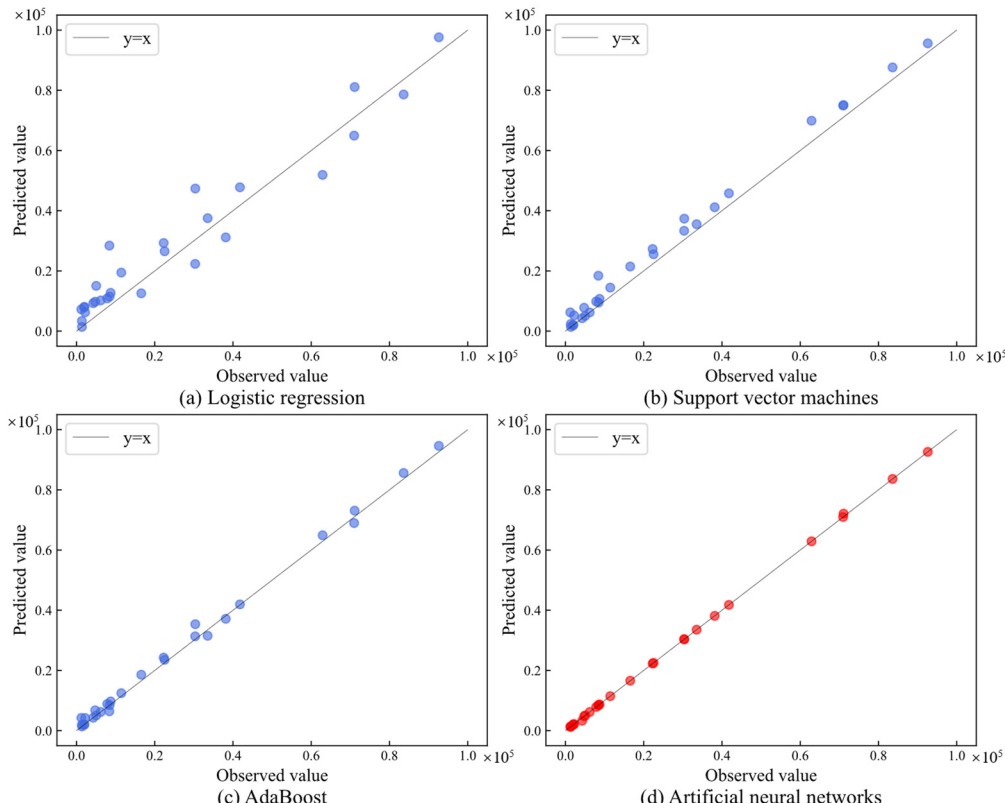

**Figure 11.** Comparison chart of prediction results of the ANN and the classical regression prediction model: (**a**) LR; (**b**) SVM; (**c**) AdaBoost; (**d**) ANN.

### 4.4. Graphical User Interface Development

In order to allow users to use the GRW-DBA data augmentation method and the ANN prediction model conveniently, a graphical user interface (Figure 12) was developed using the Python-based PyQT5 tool to realize functions such as data reading, data augmentation, model training, and prediction output.

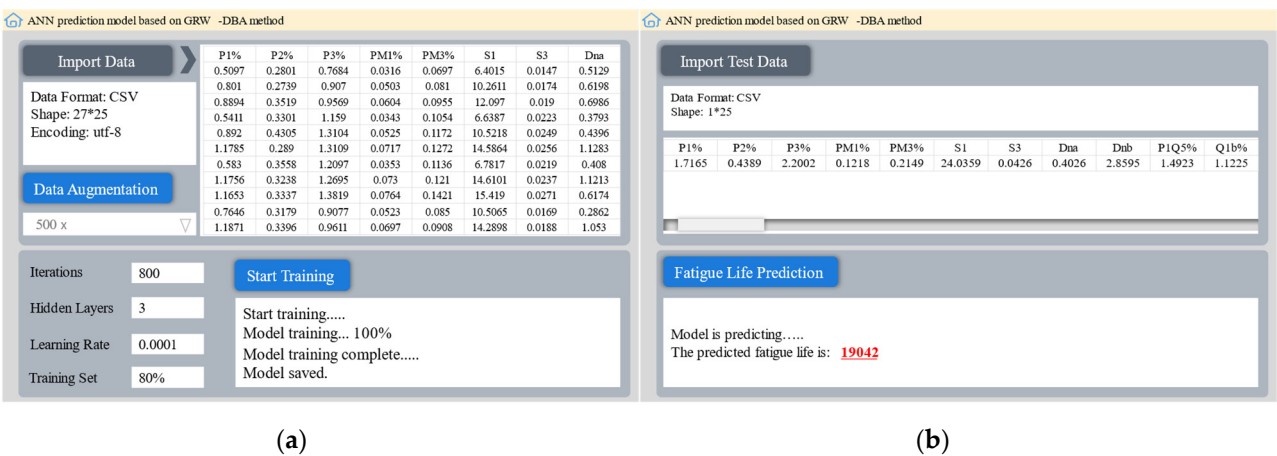

**Figure 12.** Developed graphical user interface: (**a**) interface 1; (**b**) interface 2.

The user can save the input variable as different types such as excel and text, import it into the program, and complete the data augmentation by setting the data augmentation multiple. Then, the user can set the hyperparameters of the predictive model and press the "Start Training" button to finish training the model with the augmented data. Finally, the

user presses the "Fatigue Life Prediction" button to give the prediction result according to the new data input by users.

## 5. Discussion

In this study, a new modeling approach was developed to predict the fatigue life of concrete using a small dataset. The method consists of two parts: an innovative GRW-DBA sequence data augmentation method and an ANN prediction model based on optimized hyperparameters.

The high-precision prediction of the model depends on a large amount of training data. Unfortunately, due to the limitation of experimental time and conditions, there is not enough labeled data for the task of fatigue life prediction. As an effective tool to enhance the scale and quality of training data, data augmentation is a reliable method to improve prediction accuracy. Unlike image data, in the application of sequence data, it is difficult to synthesize data and maintain correct labels due to the nature of sequence data such as inter-class correlation and large numerical differences.

Compared with data augmentation algorithms, such as GAN or classic DBA, the GRW-DBA algorithm proposed in this paper reduces the influence of outliers by grouping calculations, and at the same time, it assigns random weights in the weighted average, which can more efficiently generate data that conform to the distribution of source data. Compared with other algorithms, our algorithm can improve the accuracy of the prediction model in different concrete fatigue life data (including 27 sets and 28 sets of sequence data, respectively). Among them, the $R^2$ index increased by 10.7% and 4.2% compared with GAN and classic DBA, respectively.

Due to the data-driven and good nonlinear fitting advantages of the ANN model [49,50], it is more suitable for modeling with augmented data. Therefore, the ANN prediction model used in this paper has achieved higher prediction accuracy than classical mathematical methods such as the energy method and machine learning methods such as AdaBoost and has improved the $R^2$ index by 8.2% compared with the suboptimal AdaBoost.

Concrete fatigue damage is related to factors such as freeze–thaw cycles, shear failure, and tensile failure. One of the limitations of this study is the primary use of freeze–thaw cycle data. In the future, we will test our method on a wider dataset including other variables to improve its robustness. In addition, we will try more data augmentation algorithms, such as literature [51–53], to further improve the prediction accuracy.

## 6. Conclusions

Aiming at the problem of low prediction accuracy of concrete fatigue life under the condition of small datasets, this paper proposes a method based on the GRW-DBA data augmentation algorithm and ANN to build a prediction model in which the GRW-DBA algorithm is improved by grouping calculation and adding random weights based on the classic DBA algorithm. In this method, the ANN model is first trained based on the augmented data of the GRW-DBA algorithm, and the hyperparameters of the ANN model are optimized through experiments to improve the prediction accuracy. The data augmentation algorithm is compared with prediction models such as SVM, and finally, a graphical user interface is developed. The following conclusions can be obtained:

(1) The GRW-DBA data augmentation method proposed in this study can conveniently and effectively augment small datasets while reducing the impact of abnormal sequences on the results. Compared with GAN and classic DBA methods, the GRW-DBA can better improve the prediction accuracy of the ANN model.

(2) The ANN fatigue life prediction model was trained based on the GRW-DBA augmented dataset, under the same conditions. Its prediction accuracy $R^2$ evaluation index increased by 24%, 10.4%, and 7.8% compared with LR, SVM, and AdaBoost. It also shows good generalization in datasets with different distributions.

**Author Contributions:** Conceptualization, J.S. and Y.Z.; methodology, W.Z.; software, W.Z.; validation, J.S. and W.Z.; writing—original draft preparation, W.Z.; writing—review and editing, J.S.; supervision, Y.Z.; project administration, Y.Z. All authors have read and agreed to the published version of the manuscript.

**Funding:** This work was supported in part by the National Natural Science Foundation of China, Grant/Award Numbers: 12162025, 11762015; Natural Science Foundation of Inner Mongolia, Grant/Award Numbers: 2020MS05031, 2021LHMS05010; Basic Scientific Research Expenses Program of Universities directly under Inner Mongolia Autonomous Region, Grant/Award Numbers: JY20220139.

**Institutional Review Board Statement:** Not applicable.

**Informed Consent Statement:** Informed consent was obtained from all subjects involved in the study.

**Data Availability Statement:** All the data used in this paper can be traced back to the cited references.

**Conflicts of Interest:** The authors declare no conflict of interest.

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
