# Peer review of "ANN Prediction Model of Concrete Fatigue Life Based on GRW-DBA Data Augmentation"

_applsci, doi:10.3390/app13021227_

Round 1

Reviewer 1 Report

Overall the paper is well written and of interest. However, we note the absence of experimental data or predictions from other calculations are available for comparison. The accuracy and validity the proposed model are therefore unclear. In consequence, the author needs to address the evidence before this reviewer agrees with publication of this paper. Hence the originality and novelty of manuscript (or the proposed the methods) are not clear. This is a good paper, but you need to conduct a “Minor revision”. After those corrections the manuscript may be published in the Journal. The following comments are split into some general ones and some more specific comment. The comments are as follows:

Reviewer 2 Report

In this paper, a method is proposed to improve the prediction accuracy of the machine learning model for concrete fatigue life under small data sets, a group calculation and random weight dynamic time warping barycentric averaging (GRW-DBA) data augmentation method.

The paper’s subject could be interesting for readers of journal. Therefore, I recommend this paper for publication in this journal but before that, I have a few comments on the text that should be addressed before publication:

Comments:

1) All the abbreviations in the abstract section, and other section as well, must be explained for the first time. E.g., in line 17: GAN.

2)Which software has been used in this work to export the charts and diagrams in this work? For instance, software like SigmaPlot or SmartDraw are used to export and depict charts. Mentioning used software would be helpful to future researches and studies in the field of this article.

3)how did the authors evaluate precision of training process?

4)Which software is used in this article to model and analyze data? Moreover, which software and indices are used to compare proposed model results with other existing models? Mentioning the used software would be really useful for future works.

For instance, MATLAB and Python are highly utilized by the users to model and analyze data. Of Course there is extensive range of similar ones and it is optional to use them.

5)Novelty of present work compared to the previous works must be highlighted clearly in introduction section.

6)Since recently it has been proved that computational techniques, specifically machine learning has a numerous applications in all of engineering fields, I highly recommend the authors to add some references in this manuscript in this regard. It would be useful for the readers of journal to get familiar with the application of computational techniques in other engineering fields. I recommend the others to add all the following references, which are the newest references in this field

[1] Mozaffari, H., & Houmansadr, A. (2022). E2FL: Equal and Equitable Federated Learning. arXiv preprint arXiv:2205.10454.

[2] Roshani, et al., 2018. Density and velocity determination for single-phase flow based on radiotracer technique and neural networks. Flow Measurement and Instrumentation, 61, pp.9-14

[3]Mozaffari, H., & Houmansadr, A. (2020, January). Heterogeneous private information retrieval. In Network and Distributed Systems Security (NDSS) Symposium 2020.

[4] Zhou, Z., Davoudi, E., & Vaferi, B. (2021). Monitoring the effect of surface functionalization on the CO2 capture by graphene oxide/methyl diethanolamine nanofluids. Journal of Environmental Chemical Engineering, 9(5), 106202.

[5] Alanazi, A.K.;et al. Application of Neural Network and Time-Domain Feature Extraction Techniques for Determining Volumetric Percentages and the Type of Two Phase Flow Regimes Independent of Scale Layer Thickness. Appl. Sci. 2022, 12, 1336

Reviewer 3 Report

The paper needs English editing, especially the abstract. For instance in abstract “under small data sets” should be “using small data sets”. Also, please link the sentences in the text “Determine the optimal scaling” ..etc. Please re-write the abstract and include quantitative results.

Introduction needs revision and editing as well:

“. however” should be “. However” ..etc

The introduction and motivation behind this work should be improved. Please include more related works and motivate better the work.

There are several papers on using data augmentation “Amyar A, Ruan S, Vera P, Decazes P, Modzelewski R. RADIOGAN: deep convolutional conditional generative adversarial network to generate PET images. In2020 7th International Conference on Bioinformatics Research and Applications 2020 Sep 13 (pp. 28-33).”.

Method

The method section should be re-written. In 2.1, please link the different paragraph together or summarize them in an algorithm.

Discussion section is missing. Please include the discussion section in the paper. Authors should discuss the results and put them in the context of other works. They should also provide a limitation section and discuss alternative approaches that can be used in the future. Other methods for in-direct data augmentation can be used, such as

1- Amyar, A., Modzelewski, R., Vera, P., Morard, V. and Ruan, S., 2022. Weakly Supervised Tumor Detection in PET Using Class Response for Treatment Outcome Prediction. Journal of Imaging, 8(5), p.130.

2- Amyar A, Modzelewski R, Vera P, Morard V, Ruan S. Multi-task multi-scale learning for outcome prediction in 3D PET images. Comput Biol Med. 2022 Dec; 151(Pt A):106208.

And the impact of choice of machine learning algorithms on the results

3- Amyar, A., Guo, R., Cai, X., Assana, S., Chow, K., Rodriguez, J., Yankama, T., Cirillo, J., Pierce, P., Goddu, B. and Ngo, L., 2022. Impact of deep learning architectures on accelerated cardiac T1 mapping using MyoMapNet. NMR in Biomedicine, 35(11), p.e4794.

Round 2

Reviewer 2 Report

ready for publication in the presen form

Reviewer 3 Report

The authors have responded to all my comments. Thank you.